## Overview Review

mental health; knowledge mobilisation; low- and middle-income countries; dissemination; research impact

**Corresponding author:**
Cintia Faija;
Email: cintia.faija@liverpool.ac.uk

# Optimising knowledge mobilisation for mental health research in low- and middle-income countries: A systematic review of the state of knowledge and directions for future research

Cintia Faija[1,2], Penny Bee[2], Rebecca Pedley[2], Mia Bennion[2], Herni Susanti[3], Fitri Fausiah[4], Sri Idaiani[5], Dwie Susilo[6], Mohammad Hussen[2] and Helen Brooks[2] (ID)

[1]Department of Primary Care & Mental Health, Institute of Population Health, Faculty of Health and Life Sciences, University of Liverpool, Liverpool, UK; [2]School of Health Sciences, Division of Nursing, Midwifery and Social Work, Manchester Academic Health Science Centre, University of Manchester, Manchester, UK; [3]Mental Health Nursing Department, Faculty of Nursing Universitas Indonesia, Depok, West Java, Indonesia; [4]Research of Community Mental Health Initiatives (RoCMHI), Faculty of Psychology, Universitas Indonesia, Kota Depok, Jawa Barat, Indonesia; [5]Research Centre for Preclinical and Clinical Medicine, National Research and Innovation Agency Indonesia, Cibinong, West Java, Indonesia and [6]Department of Public Policy and Management, Faculty of Social and Political Sciences, Universitas Gadjah Mada, Yogyakarta, Indonesia

## Abstract

Optimising knowledge mobilisation in low- and middle-income countries (LMICs) could prove beneficial for improving mental health care, alleviating the global burden of high prevalence mental health problems and reducing inequalities. This review aimed to systematically review and synthesise the evidence on knowledge mobilisation for mental health improvement in LMICs, identifying barriers and facilitators and recommendations to guide progress. Four electronic databases were searched from inception to March 2024 using free text syntax combining synonyms of knowledge mobilisation, mental health and LMICs. Articles were eligible for inclusion if they were peer reviewed, on the topic of mental health and included evaluation data on knowledge mobilisation undertaken in LMICs. Included studies were quality assessed using the mixed-methods appraisal tool, and data extracted and synthesised narratively, complemented with the use of the framework for knowledge mobilisers and thematic analysis. Seventy-eight studies met the inclusion criteria. Successful knowledge mobilisation within resource-constrained settings was supported by several key facilitators. These include promoting community participation, engaging local stakeholders from the start and maintaining that engagement and building trust through equitable, long-term partnerships. Using structured frameworks helps guide systematic involvement, while fostering local ownership and leadership ensures sustainability and relevance. Knowledge mobilisation in low-resource settings faced several barriers, including limited logistical and financial resources, low literacy levels and a general lack of awareness about psychological interventions. A lack of mental health-specific training and language or translation difficulties further hindered efforts to effectively mobilise and implement mental health knowledge. Future knowledge mobilisation efforts could be strengthened by fostering sustained, trust-based collaborations among stakeholders and engaging policymakers early to ensure optimal alignment and buy-in. Emphasising local beliefs and attitudes is crucial, as is creating inclusive, participatory environments that encourage broad community involvement. Employing culturally responsive, community-driven frameworks can enhance relevance and impact, while rigorous evaluation of mobilisation strategies is critical to guide future research investment and resource allocation. Mobilising mental health knowledge in LMIC shares principles with mobilising other types of knowledge but differs in focus, stakeholders and challenges due to the stigma of mental health problems, its complexity, cultural sensitivity, misconceptions and resistance.

## Impact statement

Recent increases in global mental health funding have led to a surge in research activity across low- and middle-income countries (LMICs), improving our understanding of mental illness and informing intervention priorities. However, evidence on how best to translate this research into practice within these settings remains fragmented. Despite growing global investment, significant gaps persist in funding and in the strategic use of research to strengthen mental health systems.

While knowledge mobilisation – the process of creating, sharing and applying research to influence policy and practice – is increasingly recognised as vital, most documented strategies and frameworks have emerged from high-resource contexts. As a result, limited attention has been given to the diverse needs and capacities of LMICs, raising concerns about the generalisability and effectiveness of existing models in these settings. Addressing this gap is critical to ensure that knowledge mobilisation approaches are contextually appropriate, scalable and grounded in evidence.

This article represents the first systematic attempt to review and synthesise knowledge mobilisation practices for mental health across LMICs. The findings offer transferable, evidence-based recommendations to enhance the design, implementation and evaluation of mobilisation strategies in diverse low-resource contexts. By presenting a contemporary synthesis of the global evidence base, this work supports the development of more generalisable, locally relevant approaches that can strengthen mental health policy and practice, maximise the impact of research and contribute to sustainable improvements in mental health outcomes across LMICs.

## Introduction

There is a growing emphasis worldwide on the use of knowledge mobilisation to improve mental health systems, policies and practice. As such, the task of mobilising knowledge from research into practice has acquired increasing relevance over the last decade and is continually evolving. It is accepted as a core principle in high-quality mental health research and an important juncture for any successful research study.

In its most basic form, knowledge mobilisation is a process of creating, sharing and using research evidence (Langley et al., 2018). It is often used in the healthcare literature as an umbrella term for multiple forms of knowledge sharing activities, including but not limited to knowledge translation, exchange and dissemination. For this review, we define knowledge mobilisation as an active, iterative and collaborative two-way process between researcher and research users, for ensuring that the right evidence is applied to policy and practice (Powell et al., 2017; Ward, 2017). Research users encompass a wide range of stakeholders, including policymakers, healthcare practitioners, patients and community organisations, spanning public, professional or political levels. For instance, in mental health research, knowledge mobilisation might involve researchers working with local community leaders and health workers to understand how to improve access to culturally appropriate mental health care, or with patients and policymakers to ensure common priorities are identified and enacted at both a meso and macro level.

This definition acknowledges that engaging and working with research users can occur at any time in the research process (i.e., beginning, during and end) and encompasses activities, such as research priority setting, co-designing interventions or making decisions about how research can be best conducted effectively, efficiently and collaboratively with stakeholders. It discriminates between these active forms of information exchange, and more passive or spontaneous dissemination. Dissemination is a one-way process where research findings are communicated to relevant audiences, while knowledge impact, sometimes used as another misnomer for knowledge mobilisation, can refer to both spontaneously and strategically achieved research contributions to the world.

There are four key aspects of knowledge mobilisation: (1) to create new knowledge that is meaningful to users and to catalyse change; (2) to ensure useful knowledge reaches those that need it; (3) to warrant that knowledge is accessible and timely and (4) to close the research practice gap globally in health (Ward, 2017). Ideally, multiple forms and sources of knowledge, including patient experience, practitioner beliefs and social values, are considered in this process. This demarcates contemporary models of knowledge mobilisation from traditional evidence hierarchies, favouring only factual knowledge, and has initiated a transition from more linear routes to knowledge mobilisation to more flexible, context-driven ones.

There remains a need to ensure that knowledge mobilisation is evidence-based and continually incorporates new learning to ensure that it can optimally improve the quality, relevance and translation of research (Sanders et al., 2004; Golhasany and Harvey, 2023). Commensurate with the distribution of health research funding globally, considerable emphasis has focused on documenting, consolidating and advancing knowledge mobilisation strategies in high resource contexts, with comparatively less attention directed towards the needs and capacities of lower income settings. Knowledge mobilisation in LMICs is often constrained by contextual challenges, such as limited infrastructure, workforce shortages and weaker institutional links between researchers, policymakers and communities (Hanlon et al., 2014; Koon et al., 2020). In LMICs, knowledge mobilisation must navigate power imbalances, stigma towards mental illness, under-resourced health systems and diverse sociocultural contexts (de Leeuw et al., 2020; Vaishnav et al., 2023).

The drive to ensure health and social care services develop efficiently in a manner that is both value and evidence-based is critical to the management of long-term health conditions with a high burden of disease. In resource-constrained settings, particularly, successful knowledge mobilisation is fundamental to initiate context-relevant services, optimise impact and sustainability and reduce the time delays and economic costs of research waste.

Mental health provides an invaluable and timely lens through which to progress our understanding of knowledge mobilisation practices in resource-constrained contexts. Worldwide, ~1 billion people are currently experiencing a mental or neurological disorder, and more than 80% of these people live in LMICs (World Health Organisation, 2022). Projections of global mortality indicate that by 2030, depression is likely to be one of the three leading causes of this disease burden (Mathers and Loncar, 2006) and lack of recognition and access to care for common mental health disorders (e.g., depression and anxiety) may result in a global economic loss equivalent to a trillion US dollars each year (World Health Organisation, 2017). A subsequent systematic review has reported disproportionate impacts from the coronavirus disease 2019 pandemic in these countries, resulting in a further increase in mental health disorders in LMICs, exacerbated by new demands for care (Kola et al., 2021). In 2019, the World Health Organisation (WHO) launched a Special Initiative for Mental Health (2019–2023) to increase the accessibility of high-quality and affordable mental healthcare (World Health Organisation, 2019). Genuine stakeholder engagement in mental health system design and the application of best evidence and knowledge is critical to ensure that successful system strengthening can be achieved within these complex environments.

WHO and other international organisations actively promote learning to determine the most effective strategies to mobilise knowledge and reduce global health inequities (Welch et al., 2012). A prior scoping review has suggested that in LMIC settings, health and care policymaking most frequently draws upon scientific knowledge, usually in the form of research findings and, to a lesser extent, technical advice and routine health data (Koon et al., 2020). Capitalising efforts to enhance and understand how to best mobilise knowledge in LMICs is thus central to overcoming the challenges to research equity that LMIC researchers and research institutions face (McGregor et al., 2014; Franzen et al., 2017; Bowsher et al., 2019). Prior reviews have advanced our understanding of effective knowledge mobilisation mechanisms for public health – including the identified challenges and facilitators, the role of political and institutional influences, and effectiveness of different approaches (LaRocca et al., 2012; Oliver et al., 2014; Malla et al., 2018). These reviews offer broad-based learning, but do not describe in any detail the scope and success of the LMIC-specific practices currently being used to address the mental health treatment gap. Additionally, there is a gap in how to effectively engage diverse stakeholders – especially community members, service users and policymakers – in co-producing and using knowledge for mental health policy and practice (Hanlon et al., 2014; Ghebreyesus et al., 2022).

This review aimed to systematically review and synthesise the evidence on knowledge mobilisation for mental health improvement in LMICs, identifying barriers and facilitators, and recommendations to guide progress. We anticipate that findings from this study will contribute to optimising the utilisation of evidence in LMICs, to inform and enhance mental health policy and practice, increase access to mental health treatment and reduce mental health inequalities.

## Methods

Four electronic databases (i.e., PsycInfo, PubMed, Web of Science and CINAHL Plus) were searched from inception. The original search was conducted in October 2022 and updated in March 2024. Methods and results were reported according to Preferred Reporting Items for Systematic Reviews and Meta-Analyses (PRISMA) guidelines (Page et al., 2021). The systematic review protocol was registered in PROSPERO and can be accessed here: https://www.crd.york.ac.uk/prospero/display_record.php?ID=CRD42022330138

### Eligibility criteria

Peer-reviewed journal articles on mental health reporting on evaluation data on knowledge mobilisation with data collection taking place in an LMIC (as defined by the Organisation for Economic Cooperation Countries Development Assistance Committee [DAC] list for 2022) were eligible for inclusion. Evaluation data on knowledge mobilisation activities included studies reporting on: (1) primary empirical data and/or (2) authors's informal reflective/perspective-based accounts (i.e., no empirical data) (Ferlie et al., 2012). To facilitate reasoning for exclusion and transparency, a hierarchy flow-diagram chart was developed (Appendix 1). Any study design, including qualitative, quantitative and mixed methods, was included. As indicated in Table 1, articles published in languages other than English were included. Systematic reviews were excluded, although those relevant to the

**Table 1.** Inclusion and exclusion criteria

| Inclusion | Exclusion |
|---|---|
| Peer-reviewed journal articles published in any language. | Non-peer reviewed, such as conference papers, dissertations, literature reviews, opinion letters and books. |
| | Duplicates. |
| Studies conducted/data collected within a low- and middle-income country (as defined by OECD's DAC list for 2022). Note: Studies combining data from LMICs and high-income countries will be included if there is a clear differentiation between datasets. | Studies not conducted/data not collected in LMIC. Studies not differentiating the data of LMICs and high-income countries. |
| Studies reporting on mental health symptoms may also include prevention or well-being. | Studies not reporting on mental health. |
| | Studies focused on developmental disorders, autism, ADHD, dementia, substance abuse, conduct disorder and paraphilia |
| Studies including evaluation information about the implementation/dissemination strategies used in mental health (e.g., co-production workshops, seminars, roadshows and lecturers). This may include empirical data and/or the author's views (e.g., lessons learned, challenges and facilitators) | Studies not providing evaluation data on implementation/ dissemination challenges/ facilitators/strategies. For instance, studies listing strategies used but not providing evaluation data. |
| Studies conducted on participants of all ages. | |
| Study designs: qualitative (e.g., interviews), quantitative (e.g., cross-sectional and longitudinal) and mixed methods | Systematic reviews or protocols. |

topic were noted, reference checked and incorporated in the introduction or discussion where relevant. Table 1 outlines the inclusion and exclusion criteria in full.

### Search strategy

Search terms were organised around four areas: (1) general terms referring to knowledge mobilisation (e.g., knowledge mobilisation, translational research and knowledge translation); (2) specific terms related to knowledge mobilisation methods and strategies (e.g., collaboration, outreach, workshop and conference); (3) mental health (e.g., emotional disorder, anxiety, depression and well-being) and (4) LMICs. Search terms for each of the categories were combined using the Boolean operator "AND." A full list of the terms used for each category is included in Appendix 2.

Search results from all databases were exported to EndNote, and duplicates were automatically and manually removed. Results from EndNote were then exported to Covidence software (www.covidence.org). First, all studies were screened by two independent reviewers at the title and abstract level against the inclusion and exclusion criteria. Second, any studies deemed potentially eligible were independently screened in full text by

two reviewers. Disagreements on inclusion at any of the two stages were resolved by a third reviewer or via consensus between the two reviewers involved. Reasons for exclusion at full text were recorded in a PRISMA diagram (Figure 1).

### Data extraction

An equal number of included articles was allocated to each of the nine review team members. Each researcher completed the data extraction for the allocated studies, and all studies were reviewed by a second researcher to ensure accuracy. Data were extracted using a Microsoft Excel spreadsheet including the following information: study characteristics, sample characteristics, methods used, knowledge mobilisation information, including dissemination/strategies used, and evaluation data of those (i.e., empirical data or authors' reflections). Any disagreements at the data extraction stage were discussed with a sub-team of three members (HB, RP and PB) to reach an agreement.

### Quality assessment

To assess study quality, the mixed-methods appraisal tool (MMAT) was used (Hong et al., 2018). Quality assessment was performed by one researcher if the study included empirical data on the evaluation of knowledge mobilisation strategies. Studies reporting on the author's views, personal experiences or lessons learnt were not quality assessed due to conventional appraisal tools being designed to assess the rigour and credibility of evidence-based research. However, informal reflective/perspective-based accounts were thought to offer valuable context and insight into this topic. Due to their non-empirical nature, their contribution to the overall findings was considered supplementary to the evidence derived from empirical research.

### Data synthesis

Consistent with an integrative approach to synthesising evidence, a narrative descriptive synthesis (J. Popay et al., 2006) approach complemented with thematic analysis (Thomas and Harden, 2008) was used to analyse, integrate and synthesise the findings. The stages outlined in the Guidance on the Conduct of Narrative Synthesis Systematic Reviews were followed. First, a preliminary synthesis, including a description of the included studies, LMICs and non-LMICs involved, sample, setting, study aims, mental health focus, knowledge mobilisation strategy used and contributions (i.e., evidence, author reflections or both), was performed. Second, the framework for knowledge mobilisers (Ward, 2017), which was developed from reviewing 47 knowledge mobilisation models, was implemented for each of the included studies to increase clarity and to consistently report on Why? Whose? What? and How? knowledge is being mobilised. Each of these four questions includes accompanying categories illustrated in Appendix 3. Third, thematic analysis was applied within deductive categories, including facilitators, challenges and recommendations to mobilise mental health knowledge in LMICs.

### Role of funding source

This research was funded by the NIHR (Global Health Research for Sustainable Care for anxiety and depression in Indonesia [Award ID NIHR134638]) using the UK international development funding from the UK Government to support global health research. The views expressed in this article are those of the author(s) and not necessarily those of the NIHR or the UK government.

## Results

### Descriptive information

The PRISMA flow diagram is presented in Figure 1. Seventy-eight articles met the inclusion criteria following screening of the full text. See Appendix 4 for the full list of included studies.

Appendix 4 provides information for each of the included studies in relation to countries involved, stakeholders whose knowledge is being mobilised, setting, study aims, mental health focus, knowledge mobilisation strategy used and contributions (i.e., evidence, author reflections or both). Appendix 5 includes information on the quality assessment for each of the included studies, providing empirical data using the MMAT (Hong et al., 2018).

As shown in Appendix 4, 60 of the 78 studies were published between 2020 and 2024. Studies were conducted in different LMICs, with most of them in the region of Africa ($n = 34$), followed by Asia ($n = 23$), South America and the Caribbean ($n = 7$), Europe ($n = 2$) (i.e., Ukraine) and Central America ($n = 2$), and the remaining ones were conducted in LMICs across regions ($n = 10$). LMIC regions, including the Middle East and North Africa, Central Asia and the Pacific Islands, were either under-represented or not reported in the identified studies, highlighting a geographic gap in the current evidence base. From the included studies, 18 reported involvements of high-income countries (i.e., Australia, Canada, Germany, Ireland, the Netherlands, Norway, Russia, Spain, Switzerland, the United States and the United Kingdom). Studies conducted in primary and community health-care settings were dominant, and inpatient mental health settings were under-represented. Most of the studies refer to mental health or emotional distress in general ($n = 40$), followed by common

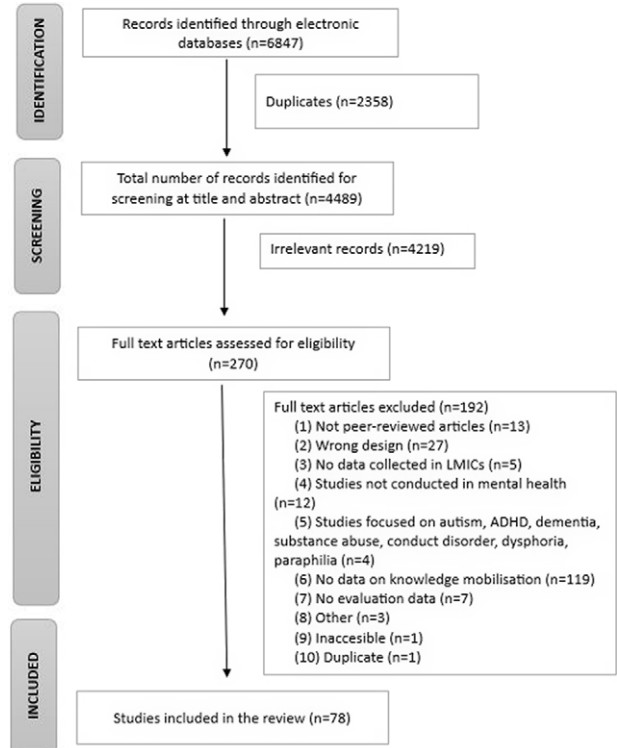

**Figure 1.** PRISMA flow diagram.

mental health disorders (i.e., depression and anxiety) (*n* = 23), and the remaining referred to a specific diagnosis (e.g., obsessive compulsive disorder, post-traumatic stress disorders, bipolar and psychosis) (*n* = 15).

### Framework for knowledge mobilisers

Appendix 6 includes information on the mapping of all included studies to the framework for knowledge mobilisers (Ward, 2017). Most of the studies were mapped into a combination of reasons to mobilise the knowledge, the two most common being to adopt/implement specific practices and to develop local solutions to practice-based problems. Regarding whose knowledge is being mobilised, many studies were mobilising knowledge from multiple stakeholder groups, including professional knowledge producers, frontline practitioners and providers responsible for delivering mental health care, service users and decision-makers. Details on multiple stakeholders per study are included in Appendix 4. Many studies were mobilising knowledge from multiple stakeholder groups, where distinct but complementary roles are played by professional knowledge producers, frontline practitioners and decision-makers. Professional knowledge producers (e.g., academic researchers and policy analysts) generate empirical evidence and theoretical frameworks that inform understanding of mental health needs and intervention effectiveness. Frontline practitioners (e.g., community health workers and clinicians) contribute contextually grounded, experiential knowledge based on direct interaction with service users and local health systems. Their insights play a critical role in understanding local realities and adapting evidence. Decision-makers in LMICs (e.g., government officials and health administrators) hold the authority to prioritise, fund and institutionalise mental health interventions. They operate within fragmented governance structures, with limited resources or political will to act on emerging evidence, and played a pivotal role in scaling and sustaining knowledge use through policy and system-level change. International bodies like the WHO and non-governmental organisations (NGOs) play a prominent role by providing technical guidance and frameworks (e.g., mhGAP) to support evidence-informed policy.

The group less engaged in mobilising knowledge across the included studies was programme developers (i.e., people responsible for designing, producing and/or implementing specific products, services and programmes). Most of the studies mobilised technical knowledge, including practical skills, experiences and expertise. Most commonly, the knowledge was mobilised by facilitating interactive learning and co-production and/or via making connections involving individuals and organisations developing partnerships and ensuring multi-stakeholder collaboration at the local, national and international levels.

### Quality appraisal

The MMAT was used to critically appraise 37 of the 78 included studies. The remaining 41 studies were not appraised as they were conceptual discussions, descriptive reports or author reflections without primary data, and thus did not meet MMAT eligibility for empirical appraisal. The 37 appraised studies included qualitative, quantitative, descriptive and mixed-methods designs. Most studies met the majority of MMAT criteria, with strong alignment between research questions, data collection methods and analysis. A small number of mixed-methods studies lacked clear integration

of findings or justification for the chosen design. Some studies had incomplete reporting (e.g., participant characteristics), which limited detailed assessment. Overall, the quality of appraised studies was judged to be acceptable and methodologically sound to support synthesis.

### Thematic analysis

Table 2 specifies studies informing each of the themes. Figure 2 provides a visual representation of the five developed themes informing on facilitators to mobilise mental health knowledge in LMICs. Similarly, Figure 3 depicts themes on challenges and recommendations to improve mental health knowledge mobilisation in LMICs.

### Facilitators of knowledge mobilisation

**Theme 1:** *Promoting community participation in mental health* **via** *an interactive, inclusive, collaborative and learning environment.*

Community participation in mental health was actively fostered through interactive and inclusive strategies, such as small group discussions, collaborative brainstorming sessions, learning groups, capacity-building workshops and reflective dialogues. Across studies, a recurring emphasis emerged on creating a collaborative and cooperative environment – one rooted in a spirit of shared learning and mutual respect. This environment was central to the success of participatory planning and implementation in mental health research across LMICs.

A key facilitator of effective engagement was the presence of balanced partnerships – where both professional experts and peer mentors contributed equally. Active listening, rather than directive instruction, was repeatedly highlighted as essential. Researchers and academics stepped away from top-down approaches, instead cultivating open, inclusive spaces where local stakeholders were encouraged to share lived experiences, express concerns and co-define priorities. For instance, Rivera et al. (2008) highlighted the value of "responding by listening" rather than acting vertically as specialists, and noted that this was possible by beginning meetings with shared dialogue (Rivera et al., 2008). Similarly, Memiah et al. (2022) illustrated how allowing local stakeholders to voice their own concerns enabled more responsive mental health interventions. Orengo-Aguayo et al. (2020)) positioned local leads as experts, promoting and affirming community ownership of the process, while Spagnolo et al. (2020)) further illustrated how structured spaces for participants to reflect on preliminary findings and offer feedback played a crucial role in contextualising results and shaping local health system improvements. In these studies, active listening was a deliberate equity strategy for fostering equitable engagement and enhancing local relevance.

Studies specifically reporting on the Theory of Change highlighted the challenges and opportunities of its use with community stakeholders. While some participants lacked confidence in leading the process, the presence of a neutral and experienced facilitator helped ensure equity and inclusiveness. Such facilitation encouraged open dialogue, respected all viewpoints and empowered all participants – academic and non-academic alike – to contribute meaningfully.

Studies focusing on cross-cultural adaptation and intervention implementation emphasised the importance of time, flexibility, patience and openness during these iterative processes. Collaborative brainstorming and ongoing reflection were essential in bridging differences in socioeconomic backgrounds

**Table 2.** Studies informing themes regarding facilitators to mobilise mental health knowledge in LMICs

| Studies | Contribution: Empirical evidence (EE) and/ or author reflections (AR) | Theme 1: *Promoting community participation* via *an interactive, inclusive, collaborative and learning environment* | Theme 2: *Negotiating initial and continuous engagement* | Theme 3: *Developing trust, equitable partnerships and long-lasting relationships* | Theme 4: *Using frameworks to facilitate systematic engagement* | Theme 5: *Fostering ownership and leadership* via *training workshops* |
|---|---|---|---|---|---|---|
| (Abayneh et al., 2020) | EE | ✓ | | | | |
| (Abayneh et al., 2022)* | EE | ✓ | | | | |
| (Abdulmalik et al., 2013) | AR | ✓ | | ✓ | | ✓ |
| (Abrahams et al., 2022) | EE | ✓ | | | | |
| (Adams et al., 2020) | EE | ✓ | ✓ | | | |
| (Afifi et al., 2011)* | AR | ✓ | | ✓ | ✓ | |
| (Agarwal et al., 2023) | EE + AR | ✓ | ✓ | ✓ | | |
| (Alonge et al., 2020) | AR | | ✓ | ✓ | | ✓ |
| (Alonzo, 2023)* | AR | | ✓ | | | ✓ |
| (Appiah et al., 2022) | AR | ✓ | | | | |
| (Atujuna et al., 2021) | AR | ✓ | | | | ✓ |
| (Ayuso-Mateos et al., 2019)* | EE | ✓ | | ✓ | | ✓ |
| (Babatunde et al., 2022) | EE + AR | ✓ | | ✓ | | |
| (Baheretibeb et al., 2022) | EE | ✓ | | | | |
| (Bitta et al., 2020)* | EE + AR | ✓ | ✓ | ✓ | | |
| (Black et al., 2023)* | AR | ✓ | ✓ | | | |
| (Brooks et al., 2021)* | AR | | ✓ | ✓ | | |
| (Brown et al., 2020)* | AR | | | ✓ | | |
| (Calia et al., 2022)* | AR | ✓ | | ✓ | | |
| (Chibanda et al., 2016)* | AR | ✓ | | | | |
| (Chumo et al., 2022)* | EE + AR | ✓ | ✓ | | | |
| (Coleman et al., 2021) | EE + AR | | ✓ | ✓ | | |
| (Demissie et al., 2024)* | AR | ✓ | | | | ✓ |
| (Fuhr et al., 2020)* | EE + AR | | | | ✓ | |
| (Fulone et al., 2019)* | EE | | | ✓ | | ✓ |
| (Gellatly et al., 2022) | EE + AR | ✓ | | | ✓ | |
| (Giebel et al., 2024) | EE | | ✓ | | | |
| (Gigaba et al., 2024) | AR | ✓ | | ✓ | | ✓ |
| (Giusto et al., 2023) | AR | ✓ | | | | ✓ |
| (Greene et al., 2022) | AR | ✓ | | ✓ | | |
| (Hafting et al., 2023) | EE | ✓ | ✓ | | | |
| (Hailemariam et al., 2015) | EE | | | ✓ | ✓ | |
| (Hamdani et al., 2021) | EE | ✓ | | | | ✓ |
| (Hameed et al., 2019) | AR | ✓ | | | | |
| (Haroz et al., 2019) | AR | ✓ | | | | |
| (Hoven et al., 2008)* | EE + AR | ✓ | | | | |
| (Kaysen et al., 2013)* | EE + AR | ✓ | | | | ✓ |
| (Khan et al., 2023) | AR | | | | ✓ | |
| (Kohrt et al., 2022)* | EE + AR | ✓ | | | ✓ | |
| (Laurenzi et al., 2024) | AR | ✓ | | | | ✓ |

(*Continued*)

**Table 2.** (*Continued*)

| Studies | Contribution: Empirical evidence (EE) and/or author reflections (AR) | Theme 1: *Promoting community participation via an interactive, inclusive, collaborative and learning environment* | Theme 2: *Negotiating initial and continuous engagement* | Theme 3: *Developing trust, equitable partnerships and long-lasting relationships* | Theme 4: *Using frameworks to facilitate systematic engagement* | Theme 5: *Fostering ownership and leadership via training workshops* |
|---|---|---|---|---|---|---|
| (Le et al., 2023)* | EE + AR | ✓ | ✓ | | | |
| (Li et al., 2023) | AR | ✓ | | | | |
| (Liem et al., 2022) | AR | ✓ | | | ✓ | |
| (Lovero et al., 2022) | AR | ✓ | | ✓ | ✓ | |
| (MacDougall et al., 2022)* | EE + AR | ✓ | | ✓ | | |
| (Maddock et al., 2023) | AR | ✓ | | | | |
| (Magidson et al., 2015)* | AR | | | | | ✓ |
| (Matsea et al., 2022) | AR | ✓ | ✓ | | ✓ | |
| (Maulik et al., 2016)* | AR | ✓ | | ✓ | | |
| (Memiah et al., 2022) | EE + AR | ✓ | | ✓ | | ✓ |
| (Mukherjee et al., 2024) | AR | | ✓ | | | |
| (Murphy et al., 2024) | EE + AR | | | ✓ | | |
| (Murray et al., 2013) | AR | ✓ | | | | ✓ |
| (Mutahi et al., 2024)* | AR | ✓ | | | | |
| (Mutiso et al., 2018) | EE | ✓ | ✓ | | ✓ | |
| (Mutiso et al., 2018)* | AR | | ✓ | | ✓ | |
| (Naslund et al., 2021)* | EE | | ✓ | | | ✓ |
| (Nguyen et al., 2023) | AR | ✓ | | | | |
| (O'Donnell et al., 2022) | AR | ✓ | ✓ | ✓ | | |
| (Orengo-Aguayo et al., 2020) | AR | ✓ | ✓ | | ✓ | |
| (Passchier et al., 2019)* | AR | ✓ | | | | |
| (Perera et al., 2020) | AR | | | ✓ | ✓ | |
| (Petersen et al., 2022) | AR | ✓ | | | | |
| (Premji et al., 2021)* | EE | ✓ | | | | |
| (Rai et al., 2023) | AR | ✓ | | | | ✓ |
| (Rivera et al., 2008)* | EE + AR | ✓ | ✓ | | | |
| (Sangraula et al., 2021) | EE + AR | | ✓ | | ✓ | ✓ |
| (Sapag et al., 2016) | EE | ✓ | | | | |
| (Shidhaye et al., 2019)* | EE | ✓ | | | | ✓ |
| (Singh et al., 2021)* | AR | | ✓ | | ✓ | |
| (Sit et al., 2021) | EE | | | | ✓ | |
| (Spagnolo et al., 2020)* | AR | ✓ | ✓ | ✓ | | |
| (Ssebunnya et al., 2021)* | EE + AR | ✓ | ✓ | | | |
| (Tinago et al., 2021) | EE | ✓ | | | | |
| (Triplett et al., 2023) | AR | ✓ | | | | ✓ |
| (van der Boor et al., 2024) | EE + AR | ✓ | | | | ✓ |
| (Wasil et al., 2020) | AR | ✓ | | | | |
| (Wieling et al., 2017)* | EE + AR | ✓ | | | | |

*Note*: Authors with an * indicate data relevant to challenges and/or recommendations.

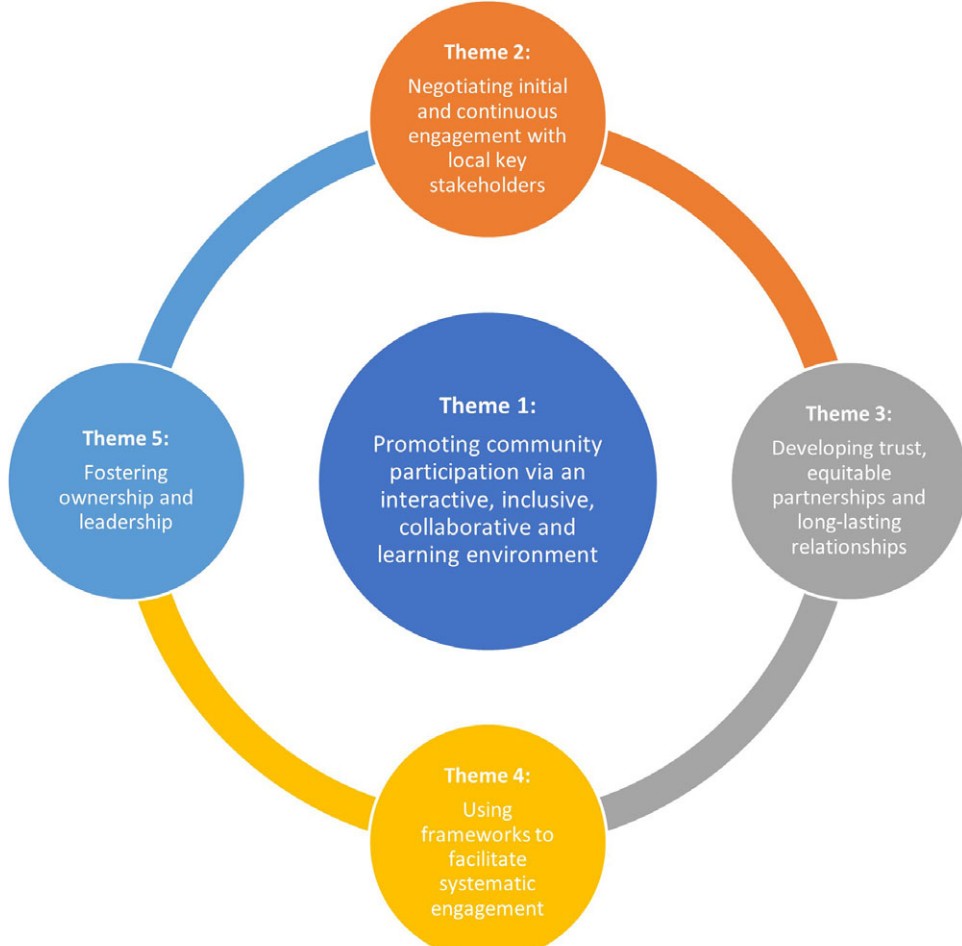

**Figure 2.** Themes informing on facilitators to mobilise mental health knowledge in LMICs.

and learning styles. Concepts, such as *reflection-in-action* and *reflective practice*, were not only useful in refining interventions but also in driving transformative change at the community level.

A limited number of studies (*n* = 2) considered the value of alternative and innovative contribution methods, such as photo-voice (Rai et al., 2023) or vignette-based approach (Haroz et al., 2019), to enable contributions from wider community audiences. In addition, Hoven et al. (2008)) found that to increase mental health awareness across different LMICs, low-cost strategies, such as various types of print media, local meetings and training sessions, were the most dominant, followed by digital media (i.e., television, radio and internet). Similarly, Ayuso-Mateos et al. (2019)) found that short courses (1–5 days) successfully built mental health capacity across (a) service users with mental health conditions and their caregivers, (b) policymakers and planners and (c) mental health researchers.

In sum, the studies under this theme highlight that authentic community engagement is most effective when it is dialogical, inclusive and sustained, grounded in local knowledge and supported by flexible, culturally attuned facilitation strategies.

**Theme 2:** *Negotiating initial and continuous engagement with different groups of local key stakeholders.*

Data highlighted the importance of identifying target communities and negotiating entry into the communities before conducting any

knowledge mobilisation activities. This was facilitated by engaging with local stakeholders and authoritative structures to get access to the system and develop an understanding of the people, their culture, beliefs, politics, power structures, dynamics and vested interests. The community members were perceived as experts who could explain their needs.

The majority of the data reported on the process of engaging with communities throughout the research and dissemination process, and the researcher dealt with gatekeepers at different levels (e.g., family, health professionals, community, cultural and political environment). Community engagement was facilitated by the identification and involvement of local mental health champions, and of people who were active mental health consultants on relevant ministries (e.g., Ministry of Health and Ministry of Education), and by regular and respectful relationships with study partners, local researchers, host communities and stakeholders. Good connections with the national government, the regional hospitals and the community health teams were of value to facilitate wider participation and commitment through the process. Similarly, working with local researchers and identifying local research procedures was identified as a potential facilitator to prevent unnecessary, lengthy processes of ethics approval.

To address power imbalances among stakeholder groups (e.g., patients, providers, community leaders, regional and national policymakers), engagement activities were conducted in varied, inclusive settings. In studies facilitating Theory of Change workshops, stakeholders contributed diverse perspectives; however, some

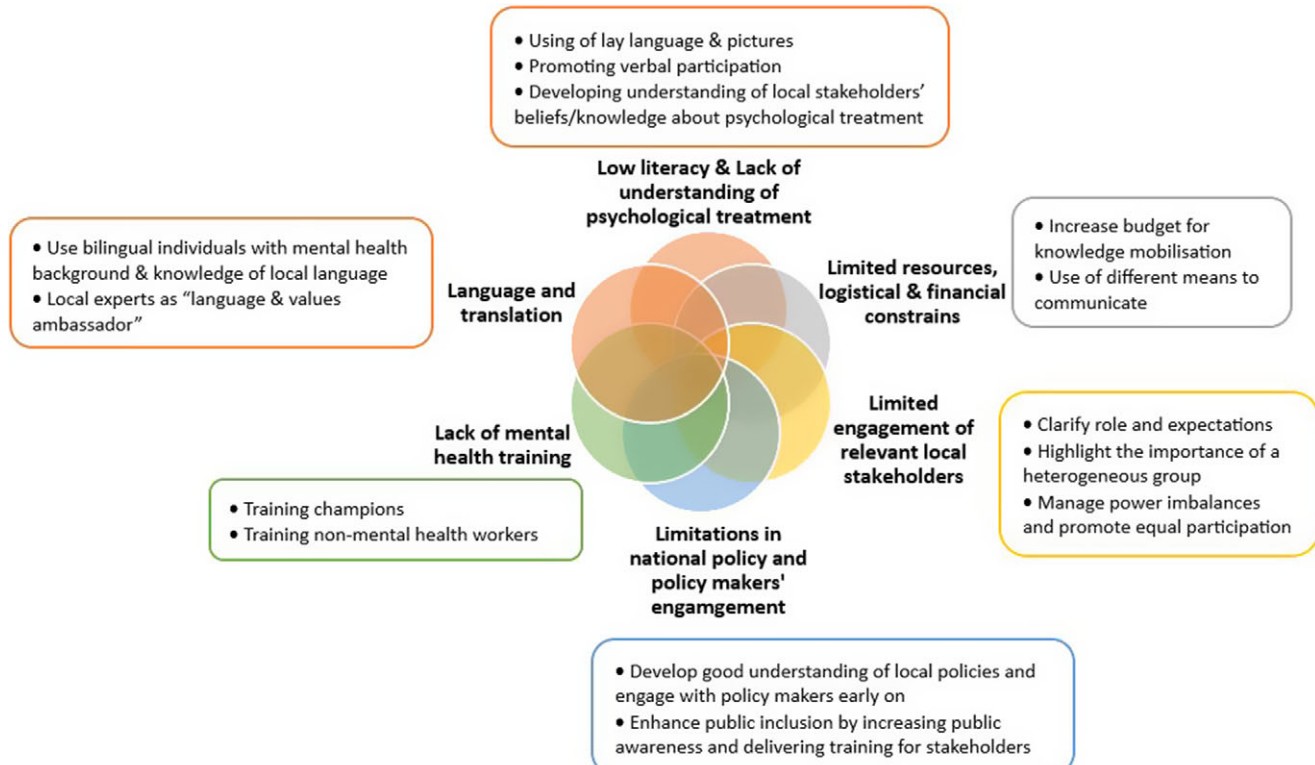

**Figure 3.** Themes informing on challenges and recommendations to improve mobilisation of mental health knowledge in LMICs.

studies favoured separate sessions to ensure equal voice, while others preferred joint sessions to promote shared understanding – although this posed challenges related to differences in education, awareness, experience, understanding, commitment and expectations. The use of visual tools in Theory of Change workshops helped bridge these gaps and improved engagement across people with different levels of education.

Studies reporting on scaling-up programmes emphasised that engagement with national-level champions, cross-sectoral teams (e.g., from health and education) and sustained policy advocacy were critical for ensuring sustained policy advocacy. Effective engagement strategies included establishing implementation teams within ministries, strategic power-sharing, joint decision-making and continual negotiation on how to align mental health priorities with broader policy agendas.

**Theme 3:** *Developing trust, equitable partnerships and long-lasting relationships.*

Findings underscored the importance of sustained, meaningful engagement with communities to foster trust, ownership and long-term collaboration. Rather than one-off involvement, ongoing partnerships with diverse stakeholders (e.g., end-users, clinicians, supervisors, patients, communities and government actors) were key to moving from passive to active participation. This consistent collaboration deepened understanding of health systems, implementation contexts and potential barriers, ensuring greater relevance and uptake of mental health interventions. Remote communication tools also supported continued engagement and efficient coordination.

In the context of cultural adaptation, early and ongoing community involvement supported iterative development, piloting and real-time responses to emerging challenges. Engaging stakeholders

in adapting materials improved their relevance and acceptability. For example, community collaboration informed anti-stigma campaigns (Maulik et al., 2016). Brown et al. (2020)) highlighted that the identification of specific sections within the materials that will need cultural adaptation, such as case stories, enabled easier adaptation in new contexts. Participatory methods, like Forum Theatre, were used to co-create content and reduce bias in cognitive interviewing (Perera et al., 2020).

Ongoing engagement with policymakers and community leaders was maintained through regular briefings and advisory boards, ensuring commitment across the development, adaptation and implementation phases. Regular communication emphasised the value of mental health research, supported resource allocation and helped sustain provider training and supervision. Multisectoral, participatory collaboration further reinforced trust, co-working and stakeholder buy-in.

**Theme 4:** *Using frameworks to facilitate systematic engagement.*

The data on the co-development and cultural adaptation processes of mental health interventions highlighted the significant role of frameworks in facilitating the exchange of ideas across diverse teams. These frameworks not only supported the collection of qualitative data on long-term sustainability and scalability but also informed iterative changes to interventions. This enabled culturally sensitive adjustments to be made before the implementation of the interventions in randomised controlled trials. More specifically, frameworks were employed to ensure the conceptual, content, semantic and technical equivalence of the adapted intervention with its original version. The use of such frameworks was perceived as adding scientific credibility and value to the co-development and co-adaptation processes.

The adaptation process typically involved situational analysis, focus groups, training workshops and the contextualization of materials tailored to specific mental health or well-being interventions. Frameworks frequently used in these processes included the Theory of Change, which articulated a vision for change and defined desired outcomes and strategies to achieve them; the RE-AIM framework (i.e., Reach, Effectiveness, Adoption, Implementation and Maintenance) (Glasgow et al., 2019), which guided intervention planning and evaluation; and the AAER framework (i.e., Adopt, Adapt, Expand and Respond) (Nippard et al., 2014), which facilitated the management and measurement of systemic change. Frameworks supporting cultural adaptation of mental health interventions included Bernal et al. (1995)), Heim and Kohrt (2019)) and the recently developed Mental Health Cultural Adaptation and Contextualization for Implementation framework (Sangraula et al., 2021), which prepares evidence-based psychological interventions for scaling in diverse contexts.

Some of these frameworks, particularly those for cultural adaptation, require specialised cultural adaptation experts and are described as "top-down" models. In contrast, frameworks like Heim and Kohrt (2019) promote an ethnographic approach, encouraging community members to serve as experts in defining, understanding and treating specific problems, which can be effectively utilised by non-specialist providers.

Frameworks like Implementation Mapping (Fernandez et al., 2019) facilitated systematic community engagement during the implementation stage. This framework provides a structured process that integrates theory, evidence and stakeholder involvement to develop implementation plans. It led to the identification of targeted strategies addressing determinants at various levels, including patient, provider, policy and community.

While these frameworks were used in various LMIC contexts, their scalability largely depends on the local adaptation and active community engagement throughout the intervention process. Their flexibility allows them to be applied across diverse settings, ensuring that mental health interventions are contextually relevant, sustainable and effective.

## Theme 5: *Fostering ownership and leadership* via *training workshops.*

Interactive and hands-on methods, such as mock sessions and role plays, were found to be effective and low-cost strategies for improving the cultural adaptation of materials before the final versions. These practical approaches, preferred over didactic lectures, encouraged early stakeholder involvement and helped promote local ownership of mental health interventions.

The use of training workshops and role plays enhanced providers' skills, built confidence in handling clinical scenarios and increased familiarity with intervention materials. They also fostered group cohesion, strengthening working relationships and enabling ongoing support and guidance.

The use of the Apprentice Model of training (Murray et al., 2011) provided a space to discuss and pilot cultural modifications during training, practice and supervision groups. This model emphasised open dialogue, local input and contextual examples – supporting adaptation while maintaining intervention fidelity and enhancing local leadership.

Hybrid training approaches, such as pre-recorded psychoeducation videos with local facilitation and virtual Q&A sessions, were deemed effective and scalable. Additional tools, like cloud-based shared documents and email communication, supported rapid, collaborative decision-making.

Cross-site capacity building – through shared resources like manuals, training slides and evaluation tools – significantly shortened adaptation timelines (e.g., from 1 year to 4 months) (Coleman et al., 2021). Establishing a Technical Working Group of local stakeholders further streamlined this process by reviewing training materials for cultural relevance and identifying necessary adaptations.

## *Challenges and recommendations to improve knowledge mobilisation*

Data across studies reporting on challenges and recommendations on mental health knowledge mobilisation in LMICs were more limited compared to the data on facilitators. See Figure 3 for a visual representation of these findings.

**Theme 1:** *Limited resources, logistical and financial constraints.*

Data analysis highlighted that the lack of funding was a common challenge across studies, which influenced stakeholders' involvement throughout the process (e.g., costs and transportation of involvement), the capacity to train all relevant groups of stakeholders involved (e.g., clinicians and health planners) and limited dissemination of the research findings in different settings.

Data across studies highlighted logistical problems (e.g., meeting location and times) and poor online connection, which increased difficulties in understanding the chosen shared language (i.e., English rather than the local language), reducing effective communication. The chat function was used to overcome poor connection, and when the connection was extremely poor, email exchange was used instead.

**Theme 2:** *Low literacy and lack of understanding of psychological interventions.*

Data analysis revealed that low literacy was one of the main challenges encountered during the participation process (e.g., group activities, forums and workshops) and the co-development of intervention materials to be culturally and context-specific. Written materials were simplified by using lay language and pictures, including training materials to aid understanding from intervention facilitators, and resources for end-users. For instance, a mood questionnaire replaced numerical ratings with facial expressions to facilitate completion for individuals who may struggle with reading numbers.

Participation in group activities involving reading and writing was also challenging due to low literacy. To ensure adequate involvement across participants, techniques promoting verbal participation and reflection were used instead.

Data on challenges faced when delivering a psychological intervention in rural communities indicated a lack of awareness and understanding of the difference between pharmacological and psychological treatment, which was attempted to be overcome by clarifying the differences between treatment by pills and treatment by talk.

Data analysis underscored the idea of evidence-based practice as this was perceived as uncommon in LMICs, which is why providing research evidence supporting a mental health intervention was not compelling to obtain buy-in from local stakeholders. Thus, buy-in was ultimately gained once the mental health intervention/programme was implemented, as local stakeholders could notice the impact on the end-users.

**Theme 3:** *Limitations in national policy and policymakers' engagement.*

Data analysis highlighted the lack of connection between science, policy and practice uptake, suggesting early engagement with policymakers via Theory of Change workshops, focus groups, community discussions and key informant interviews. In addition, keeping regular contact via meetings with policymakers throughout the process was recommended to identify the impact of research on policy and practice, the need and value for mental health research and to own and be part of the process, the findings, the practice and the policy recommendations. Data emphasised the importance of discussing the national- and county-level policies, how they are developed and implemented and exploring pathways to address policy changes and support the delivery of evidence-based treatments for mental health and promote its sustainability.

Policy limitations affected supplies and the availability of specialised human resources. Furthermore, embedding service users in primary healthcare was challenging due to a national regulatory policy enforcing this involvement.

**Theme 4:** *Limited engagement of relevant local stakeholders.*

Data analysis emphasised the lack of initial and/or continuous involvement of relevant groups of key stakeholders. In addition, stakeholder involvement was not heterogeneous (e.g., traditional healers and refugees) and participation was influenced by levels of experiences of those involved and lack of clarity on roles and expectations. One study highlighted that engagement with local stakeholders was meant to be increased by employing incentive workers with different backgrounds, but authors reported challenges with capacity and retention of short-term staff, and national restrictions on refugees in paid employment (Singh et al., 2021).

Data suggest that public awareness and training for multiple stakeholders (e.g., health researchers, clinical leads and NGOs) would be beneficial to enhance public inclusion in mental health research and intervention design.

**Theme 5:** *Lack of mental health training.*

Data analysis underlined mental health-specific training for non-specialist training of providers as a common challenge for the cultural adaptation process and co-development of evidence-based psychological interventions. Data on recommendations include the training of champions and non-mental health workers. In addition, poor communication and unclear pathways of referral between different services made the research process difficult, and tracking referrals, introducing standard operating procedures and transparent information sharing were recommended.

**Theme 6:** *Language and translation.*

Data from studies reporting on cultural adaptation processes highlighted challenges in translation and the use of different dialects. Recommendations to overcome these included incorporating bilingual individuals with both a strong mental health background and robust knowledge of local languages, to adapt terminology to be understood by different groups of stakeholders involved in these diverse communities and/or using a local expert as a "language and values ambassador" to facilitate and guide through the different steps of the cultural process in a relevant and effective manner.

## Discussion

This article represents the first attempt to systematically review and synthesise the scope and success of mental health knowledge mobilisation practices in LMICs. Previous research in this field has focused on high-income settings to the detriment of LMIC-specific contexts. This exclusive focus on high-income countries excludes LMIC perspectives and hinders the development of locally relevant mental health strategies, which are likely to be fundamental to reducing health inequalities and optimising resource allocation (Sanders et al., 2004).

The growing body of literature identified within our review highlights the increasing global interest in knowledge mobilisation in LMICs, particularly in African and Asian regions. Despite this, important gaps remain. This includes geographic gaps within regions, such as the Middle East, Central Asia and Pacific Islands, which were markedly underrepresented within included studies. Across LMIC contexts, robust evaluations of knowledge mobilisation activities were distinctly lacking, reflecting global concerns and priorities for the field (Golhasany and Harvey, 2023).

The majority of activities reported to date have taken place in primary and community settings with a focus on general mental health or common mental health problems. Knowledge gaps exist in inpatient settings and more severe forms of mental illness. This may reflect resource constraints and health system priorities in LMICs and the higher levels of stigma associated with these disorders, (Vaishnav et al., 2023), two conditions that also that also underscore the importance of achieving effective knowledge mobilisation in these settings. The combination of narrative synthesis with an existing knowledge mobilisation framework has identified that existing efforts typically involve multi-stakeholder collaboration – integrating the perspectives of professional knowledge producers, frontline practitioners, service users and less frequently decision-makers. Programme developers are notably less engaged. This reflects a contemporary and evolving commitment to participatory research and co-production in global mental health; an earlier scoping review on the same topic only identified the inclusion of professional knowledge producers in these contexts (Koon et al., 2020).

Facilitators of successful knowledge mobilisation across studies included in the current review were authentic community participation, sustained and trust-based partnerships, strategic engagement of key stakeholders, the use of existing structured frameworks and investment in local leadership and capacity to support mobilisation activities. Challenges included resource constraints, low literature, insufficient involvement and alignment to policy and inconsistent stakeholder involvement (de Leeuw et al., 2020).

General guidelines for effective knowledge mobilisation in high-income contexts have been described in the literature across different countries (e.g., the Social Sciences and Humanities Research Council in Canada, the UK National Institute for Health Research and the Australian Prevention Partnership Centre). These guidelines highlight the importance of engaging in a collaborative process between researchers and communities underpinned by trust and relationship-building. The current knowledge mobilisation literatures underscore the importance of:

1 Co-creation, collaboration and co-production with communities by engaging with stakeholders (e.g., clinicians, patients, policymakers and researchers).

2 Attention to contextual relevance to fit social, political, environmental, technological, legal and economic factors affecting the proposed research.
3 Enhanced communication by discussing research using appropriate and understandable language for different audiences.
4 Using a knowledge mobilisation theory, framework or model to plan knowledge mobilisation strategies systematically.
5 Building knowledge exchange contacts and developing relationships.

Our findings align with but also extend this guidance, indicating that mobilising mental health knowledge in LMICs requires additional consideration and strategy tailoring, due to nuanced differences in financial, human and institutional resources, infrastructure, health system priorities, cultural context and governance. For example, our synthesis shows that collaboration with communities was often inconsistently applied, with certain groups (programme developers, spiritual healers and refugees) excluded who could meaningfully contribute to knowledge mobilisation activities (Green and Colucci, 2020). Our results suggest a need for tailored engagement strategies for these groups. Some included studies used creative approaches to support engagement activities (e.g., photovoice), but substantial structural and contextual barriers remained, which required improvisation rather than systematic planning. Our findings highlight not only the value but also a central need for local adaptation of existing knowledge mobilisation frameworks to ensure optimal use in LMIC contexts.

By focusing on mental health knowledge mobilisation, our review has not only highlighted overlapping principles with other health-related fields (Haynes et al., 2020) but also raised additional considerations that need to be considered due to the unique characteristics of mental health contexts. These include:

1 The stigma of mental health problems, the cultural sensitivity, misconceptions and resistance, which require additional efforts to destigmatise mental illness, and are heavily dependent on local advocacy where champions within the community play a central role in mobilising knowledge.
2 Limited awareness and understanding of mental health problems, which prevents people from optimally engaging in mental health research and from seeking care.
3 The complexity of mental health problems, and the different needs across mental health diagnoses, requires a focus on personalised and patient-centred approaches.

We have developed the following recommendations from our findings, supported by the wider literature, to support future mental health knowledge mobilisation in LMIC contexts.

## Recommendations

### 1. Fostering personalised collaborations built on trust, active engagement and equitable partnerships between international researchers and local stakeholders.

Compared to mobilising knowledge in HICs and mobilising knowledge on any other public health topic, mobilising knowledge on mental health in LMICs requires a more intense and personalised collaboration between international researchers and local stakeholders to build trust, genuine and active long-lasting relationships. This is suggested to be related to the stigma and cultural sensitivity

nature of the topic (Sartorius, 2007) and power dynamics between stakeholders and LMIC and HIC researchers/academics, meaning that more time, effort and budget are needed to this end.

Mobilising mental health knowledge in LMICs is not about an ad hoc one-time activity or involvement with the community to culturally adapt a mental health intervention; it is a continuous, consistent and iterative process that requires time and commitment from everyone involved, where ongoing and genuine engagement is crucial to develop strong and trusty relationships, to work collaboratively and in equitable partnership, fostering ownership and leadership.

Cultural sensitivity around mental illness requires mental health interventions to be individually tailored to local cultural and religious beliefs to increase acceptance. Stakeholder engagement in LMICs is community-driven rather than professional-led, heavily relying on local leaders, community health workers, NGOs or individuals with lived experience to advocate for mental health care and ensure relevance and buy-in. Thus, a clear plan, sufficient time and budget to engage with different groups of key stakeholders is crucial, including the identification of stakeholders that are already trusted in communities, like spiritual/religious leaders (Green and Colucci, 2020).

### 2. Building early, symbiotic relationships with local policymakers to facilitate the integration of mental health strategies

Lack of mental health policy, structured networks and advisory boards promoting community engagement in mental health research in LMICs makes knowledge mobilisation challenging compared to HICs. In addition, evidence to update policies in LMICs can be slowed by bureaucracy and/or political instability.

Global mental health researchers should establish symbiotic relationships with LMICs' local policymakers from the start of the research process. Thus, time and effort should be prioritised to explore pathways to policy changes, increase awareness of mental health and encourage community involvement in mental health research. Good connections with the local government were found to be of value to facilitate wider participation and commitment through the process. In addition, ongoing engagement with policymakers via regular briefings or by developing a Community Advisory Board Project, champions were identified as fruitful in this review to mobilise mental health knowledge. It is evidenced in a recent scoping review that knowledge mobilisation processes can improve health outcomes and enhance policymaking in LMICs' health systems (Koon et al., 2020).

### 3. Promoting mental health literacy by addressing knowledge gaps and improving attitudes towards mental health

Preceding the cultural adaptation of a mental health intervention, it is key for international researchers to work in collaboration with local stakeholders to understand knowledge, attitudes and beliefs about mental health in LMICs and treatments available (Bitta et al., 2020). Evidence highlights these are key barriers deterring people from seeking help (Gulliver et al., 2010; Andrade et al., 2014) and consequently impacting on effective mobilisation of mental health knowledge in LMICs.

Limited awareness and understanding of mental health problems in LMICs makes mobilising this knowledge more challenging. Meaning that increasing awareness of mental health should be prioritised over adapting mental health evidence-based interventions (Gulliver et al., 2010). Lack of understanding and awareness of mental health problems hinders uptake on both engagement with

research on this area and more broadly uptake of psychological interventions.

Findings from this review suggest that the use of low-cost methods, such as print media, local meetings and training sessions, was the most dominant, followed by digital media (television, radio and internet) to increase mental health awareness in LMICs. Importantly, digital media and mobile mental health platforms may offer a scalable solution for knowledge mobilisation in LMICs, where access to mental health services is often limited. Mobile apps and SMS-based interventions enable the delivery of evidence-based mental health support (e.g., Cognitive Behavioural Therapy and psychoeducation), transcending geographical, resource barriers and helping reduce stigma by normalising mental health conversations. However, despite the potential, challenges such as digital literacy, internet accessibility and data privacy concerns need to be addressed to maximise the impact of these technologies in promoting mental health awareness and mobilising knowledge in resource-constrained settings (Lund et al., 2018).

Using simple language and pictures in written mental health materials, and verbal participation (rather than written participation) to engage with end-users in mental health research was emphasised to effectively mobilise knowledge, as literacy was deemed low. For instance, using "treatment by talking" as opposed to "psychological treatment" facilitated understanding and differentiation from medical treatment (e.g., taking tablets). This is particularly salient in LMICs, as there is limited awareness of alternative treatments for mental health outside medication.

Findings suggest that training for multiple stakeholders (e.g., health researchers, clinical leads and NGOs) would be beneficial to enhance public awareness and inclusion in mental health research and intervention design.

### 4. Encouraging community participation through an interactive, inclusive and collaborative learning environment

Mobilising knowledge on mental health interventions requires accounting for sensitivity, being culturally appropriate and being context-specific for the local community. A collaborative and interactive approach bringing people together as active and equal partners, valuing all types of knowledge (policymakers, providers and end-users), using interactive techniques, such as small group discussions, capacity-building workshops, reflective sessions and collaborative brainstorming, was highlighted as the most beneficial. Other more innovative methods included photovoice, vignette-based approach, forum theatre or theatre of the oppressed. The use of the Apprentice Model of training, including constant and collaborative reflection, facilitated the learning process despite differences in socio-economic conditions and learning style.

### 5. Applying comprehensive and collaborative frameworks to guide the cultural adaptation of mental health interventions

LMICs have a wide range of diverse and complex needs, including greater diversity in cultural, literacy and socio-economic status, requiring context-specific adaptations of mental health interventions where knowledge mobilisation must integrate with traditional and informal systems (e.g., traditional healers and community-based systems). This contrasts with HICs, where established evidence hierarchies (e.g., randomised controlled trials) often dominate decision-making.

Findings from this review emphasise the importance of using frameworks developed to support cultural adaptations for mental health interventions, specifically as they account for information

relevant to this subject area and highlight engagement with the local community as a key for a meaningful adaptation. Thus, the use of frameworks identifying local communities as experts rather than those using a "top-down" adaptation model is encouraged to produce and mobilise mental health knowledge. This is aligned with the recently proposed framework for delivery of comprehensive, collaborative and community-based care to expand mental health services in LMICs (Bolton et al., 2023).

### 6. Implementing robust evaluation methods and optimising limited budgets for effective knowledge mobilisation

Robust evaluation of knowledge mobilisation activities will develop our understanding of what works, what does not, when and for whom, to support future research and optimise limited budgets. Budget constraints allocated to mobilise knowledge were shared across health disciplines (Murunga et al., 2020).

Given the importance, the time and resources that are spent on knowledge mobilisation strategies and activities, it is crucial to have formal evaluations on how effective they have been in meeting the original goals for mobilising the knowledge. In addition, informal evaluations, including reflections and lessons learnt, are of value. Evaluation should also consider how the knowledge mobilisation strategy/activity has influenced others (e.g., stakeholders, practitioners, policymakers, service users and researchers). Ongoing evaluation through the life cycle of the project should inform future planning and/or adjustments to the planning.

In sum, findings from this review carry important implications for multiple stakeholders involved in mental health knowledge mobilisation in LMICs. For researchers, the need to prioritise equitable, trust-based and enduring partnerships with local communities and stakeholders is paramount to ensure contextual relevance and sustainability. Policymakers are encouraged to invest in mental health infrastructure, develop inclusive policies and establish systematic mechanisms for evidence engagement. For practitioners, accessible and culturally responsive training is essential, particularly in settings with low literacy and linguistic diversity. Finally, strengthening community participation and leadership can foster local ownership, reduce stigma and enhance the development, adoption and effectiveness of mental health interventions in resource-limited contexts.

### Strengths and limitations

This systematic review benefited from the engagement of a multi-disciplinary team including four members from LMICs, three of whom were trained mental health clinicians, and the involvement of independent reviewers throughout the screening process, ensuring thoroughness, reliability and validity of the results.

Several limitations warrant discussion. First, the lack of consistency and the wide range of terms used to refer to knowledge mobilisation proved challenging and may have influenced to adequately capture all relevant articles in the topic. Even though non-English studies were included, the use of predominantly English-language databases may have excluded relevant non-English literature.

We intended to formally evaluate knowledge mobilisation strategies and their outcomes, but these were not easily identifiable. Future research, adopting a commonly agreed definition of knowledge mobilisation and these criteria, is urgently needed. While our synthesis points to a need for research expansion, the quality of the studies meeting our inclusion criteria and providing empirical evidence had limitations. Included studies also typically focused

on the experience of undertaking knowledge mobilisation, so there were limited opportunities to make policy recommendations.

## Conclusion

This review identified a range of strategies used to mobilise mental health knowledge in LMICs, with community participation, sustained stakeholder engagement and equitable partnerships emerging as key facilitators. However, systemic barriers, such as resource limitations, low mental health literacy and fragmented policy engagement, continue to hinder progress. Addressing these challenges requires contextually grounded approaches that prioritise local leadership, strengthen cross-sector collaboration and embed knowledge mobilisation within existing structures. The findings offer actionable recommendations to support the development of more inclusive and effective mental health policies and practices, ultimately advancing equitable mental health systems in resource-constrained settings. Compared to HICs, mobilising mental health knowledge in LMICs requires more time, flexibility, patience, openness from everyone involved and additional efforts to destigmatise mental illness, and it is heavily dependent on local advocacy where champions within the community play a central role. Stakeholder and community engagement to mobilise mental health knowledge in LMICs requires a more intense, continuous and personalised collaboration.

**Open peer review.** To view the open peer review materials for this article, please visit http://doi.org/10.1017/gmh.2025.10059.

**Supplementary material.** The supplementary material for this article can be found at http://doi.org/10.1017/gmh.2025.10059.

**Data availability statement.** All relevant data are within the article and its Supporting Information files.

**Acknowledgements.** The authors would like to thank Miss Saher Nawaz for the development of the graphical abstract.

**Author contribution.** Cintia Faija: Conceptualisation; data screening, data extraction, quality appraisal for studies meeting inclusion criteria from the original search; formal analysis; methodology; project administration; writing – original draft; writing – review and editing. Penny Bee: Conceptualisation; funding acquisition; formal analysis; methodology; project administration; writing – original draft; writing – review and editing. Rebecca Pedley: Data screening, data extraction, quality appraisal for studies meeting inclusion criteria from the original search; writing – review and editing. Mia Bennion: Conducting the updated search; data screening, data extraction, and quality appraisal for studies meeting inclusion criteria from the updated search, writing – review and editing. Herni Susanti: Data screening, data extraction, quality appraisal for studies meeting inclusion criteria from the original search; writing – review and editing. Fitri Fausiah: Data screening, data extraction, quality appraisal for studies meeting inclusion criteria from the original search; writing – review and editing. Sri Idaiani: Data screening, data extraction, quality appraisal for studies meeting inclusion criteria from the original search; writing – review and editing. Dwidjo Susilo: Data screening, data extraction, quality appraisal for studies meeting inclusion criteria from the original search; writing – review and editing. Mohammad Hussen: Data screening, data extraction, quality appraisal for studies meeting inclusion criteria from the original search; writing – review and editing. Helen Brooks: Conceptualisation; data screening, data extraction, quality appraisal for studies meeting inclusion criteria from the original and also the updated search; formal analysis; funding acquisition; methodology; project administration; supervision; writing – original draft; writing – review and editing.

**Financial support.** This research was funded by the NIHR (Global Health Research for Sustainable Care for anxiety and depression in Indonesia [Award ID NIHR134638]) using UK international development funding from the UK Government to support global health research. The views expressed in this article are those of the author(s) and not necessarily those of the NIHR or the UK government.

**Competing interests.** The authors declare none.

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
