## [Reviewer Report]

While the introduction explains why this review is necessary, it could be improved by focusing more on the specific gaps in existing research on knowledge mobilisation in LMICs. This would make it even clearer why this review is so important.

Understanding Knowledge Mobilisation in Context: The definition of knowledge mobilisation is comprehensive, but a brief mention of how it plays out differently in high- versus low-resource settings could help clarify the unique challenges LMICs face.

Making the Aim Clearer: The aim of the study is precise, but tightening it up a bit could make the focus even sharper. For example, “This review aims to identify the most widely used knowledge mobilisation strategies in LMICs, explore the barriers and facilitators to their success, and provide recommendations to help improve mental health policies and practices in these regions.” This phrasing makes it feel a bit more actionable.

Clarifying the Exclusion of Non-Empirical Studies: It makes sense to prioritise studies with empirical data, but it would be helpful to briefly explain why studies based on author reflections are given less weight in the analysis. A short note on how this decision might impact the overall understanding of knowledge mobilisation in LMICs would add clarity.

Refining the Search Terms: The search strategy is thorough, but it could be helpful to specify whether terms related to policy implementation, community-based interventions, or system strengthening (in the context of mental health) were included. Highlighting these areas might help capture more research focusing on real-world knowledge mobilisation applications in LMICs.

Addressing Potential Conflicts of Interest: If any authors have affiliations or ties that could influence the findings, a brief disclosure would add another layer of transparency. Even if there are no conflicts, explicitly stating this reassures readers that the research is unbiased.

---

## [Reviewer Report]

Abstract

1. The abstract format is up to standard

2. The abstract has smooth, consistent and study objecAve-driven flow of discussion

IntroducCon

1. Well done, well referenced and well researched

Methods

1. Well done

Results

1. There is a lot of informaAon and outcomes/results in themes, which need further

collapsing to reduce the size of the document. This will move the level of data analysis

from descripAve to analyAcal, which will strengthen discussion, conclusion and

recommendaAons

Discussion

1. It needs some revision, largely because of informaAon overload (raw and descripAve) in

the results

2. Discussion is not done through subsecAon – the author should revise and make it a prose

3. Efforts should be made to reflect the enAre scope of findings in the discussion

RecommendaCons

1. RecommendaAons do not require explanaAons/discussions. ExplanaAons/discussions out

to have been done in the results and discussion secAons

2. RecommendaAons need to be reshaped to be objecAve, research quesAon and hypothesis

focused

Conclusion

1. The author should try to ensure that the conclusion is a response to the study quesAons,

objecAve and hypothesis

General comments

1. The study was well designed and conducted, most likely by a senior and experienced

researcher

2. The study needs very liUle correcAons, may be the final review of language 9sAll minimal

changes)

3. This is a high value study that warrants to be

---

## [Reviewer Report]

I have read this article several times, and it is indeed informative and relevant. While using the MMAT instrument is a good way of pooling and analyzing the work done in this area, it would have added value to readers to have some descriptive statistics to bring out a clearer picture of significance of the finding. It does not suffice to report items as “majority, few, minority etc” if you can calculate a figure. This would then inform the conclusions in a much better way.

Otherwise i must agree that this is a relevant article, well done but can be improved

---

## [Editor Report]

Comments for the editor

The manuscript addresses a critical gap in global mental health by focusing on knowledge mobilization in LMICs, where resource constraints and stigma disproportionately impact care. The use of PRISMA guidelines, PROSPERO registration, and systematic methodology enhances credibility. The discussion presents actionable strategies, including fostering equitable partnerships and utilizing culturally adaptive frameworks, which are grounded in the findings. However, the authors can address the following minor revisions before the manuscript is accepted.

Comments for the author

Abstract

The conclusion could explicitly link findings to actionable recommendations (e.g., community-driven frameworks) to strengthen policy relevance.

Introduction

Page 4, Lines 17–20: “…working with research users can occur at any time in the research process…

Please clarify what is meant by “research users” in this context. Are they policymakers, practitioners, patients, or community groups? Defining and adding a clear example would improve accessibility for broader audiences.

Methods

Page 9, Lines 23–26: “Studies reporting on author’s views, experiences or lessons learnt were not assessed for quality and weighted less heavily in the synthesis due to the nature of the data (i.e., evidence coming from personal reflections and interpretation differently to empirical data).

Justify the Exclusion of Quality Assessment for Certain Studies”. The rationale for excluding these studies from quality assessment is valid but could be expanded. Consider adding a sentence to clarify how these were weighted “less heavily” in the synthesis was it through exclusion from meta-analysis, thematic separation, or another method? This would enhance the transparency of the analytical approach.

Under eligibility criteria: … “Evaluation data on dissemination or implementation strategies included primary empirical data but also author’s report on the views, experiences or lessons learnt (Ferlie et al. 2012).” This sentence blurs the line between empirical data and personal reflections. To improve clarity, consider distinguishing more clearly between the types of data being accepted. For example, define whether “views and experiences” must be systematically collected or whether informal reflections qualify. This will help readers assess the rigor of the included studies.

It is necessary to clarify how MMAT was applied to mixed-methods studies (e.g., weighting qualitative vs. quantitative components).

Additionally, there appears to be a language bias. While the paper claims to include non-English studies, the predominance of English-language databases (e.g., PsycInfo, PubMed) may exclude relevant non-English literature. This should be explicitly addressed.

Results

Under Theme 4: Using frameworks to facilitate systematic engagement: While frameworks like mhCACI are mentioned, the paper could elaborate on how these were applied across diverse LMIC contexts and their scalability.

In the descriptive results, it might be interesting to note if there were any significant gaps in terms of specific LMIC regions or types of mental health conditions studied.

Page 12, Lines 6–8: “…many studies were mobilising knowledge from multiple stakeholder groups, including professional knowledge producers, frontline practitioners, and providers responsible for delivering mental health care, service users, and decision makers.” This sentence lists stakeholder groups but could benefit from clarification. Consider differentiating the roles of “professional knowledge producers” vs. “frontline practitioners” and “decision makers” to help readers understand their unique contributions in the knowledge mobilisation process.

Page 13, Lines 11–14: “Active community facilitation involving a balanced contribution from professional experts and peer mentors, was perceived as a facilitator to community engagement success. Good listening skills (rather than telling) from scientists and academics was identified as a fruitful way to solve dissemination and implementation challenges in LMICs.” This is an insightful observation, but you might enhance its impact by providing a concrete example or citation from the included studies that illustrates this dynamic in practice. A specific case would deepen the reader’s understanding of how “listening” was operationalized as a strategy in real settings.

Discussion

In the results under theme 2 and also in the discussion under recommendation 3, digital media is noted as a facilitator, but there is limited discussion on leveraging technology (e.g., mobile health platforms) for scalable KM.

Additionally, it would be valuable to explore the implications of the findings for different stakeholders (e.g., researchers, policymakers, practitioners, and community members).

Implementation Barriers: The “challenges” section could be expanded to address systemic issues like corruption or infrastructural deficits, which indirectly affect KM.

---

## [Reviewer Report]

Abstract

1. Non-conformity with standard practice is noted in some sections

2. Results need further synthesis to reflect a response to the study objective/statement of the problem, tied to the conclusion and research question.

3. The conclusion in particular is less consistent with standard practice, though it carries relevant content

Impact statement

1. The statement needs to be aligned to the generalizability of the findings

2. Some more refinement and inputs are required to strengthen the paper

3. Part of the discussions in this section can find relevance in the results section

Introduction

1. There is need for only one definition of knowledge mobilization to be quoted, with precision

2. Good content, which requires a cut-down on verbiage to inspire reading with smooth flow

3. Admirable simplicity of language for communication across all levels of literacy

4. There is need to reorganize introduction, starting with definition, then magnitude of the problem ……

5. There is need for more focus and intensification on the discussion/argument on knowledge mobilization more than mental problems

6. There is need for the aim of the study as stated therein, to be aligned to the study title, driven by focused literature in the paper

Methods

1. The methods section is well done, but it requires review and refinement

2. The PRISMA is cited but the flow chart to demonstrate inclusion and exclusion criteria ia missing

3. Refine and create clarity and precision in this section to make it compact

4. Demonstrate disaggregation of selected articles (by inclusion criteria) into countries and or regions

5. Good results - some further synthesis and analysis can enrich the valuable findings, unless the intention is to stop at descriptive results only

6. A lot of good findings on action but little on policy change/implementation

Discussion

1. A bit deficiency in this section, due to less analytical data management but more descriptive - a consideration for analytical level may give us great and valuable findings/discussion

2. Discussion is about the findings compared to the available literature and more so that used in the introduction/literature search - revisit, review and revise this section

Recommendations

1. This section requires refinement and clarity

2. Align the recommendations to the findings and the relevance/justification of the study

General

1. The paper is large in size and a review on the economy of words can help and strengthen the value therein.

2. In between the discussion and recommendations, there is a feeling of something missing that would fill the gap of whether the student addressed the research question or not

3 a study of immense value and relevance

---

## [Reviewer Report]

The authors have made great revisions to the manuscript. It has been refined well and should now be published.

---

## [Reviewer Report]

Cambridge Prisms: Global Mental Health

GMH-2025-0046.R2

COMMENTS

9 August 2025

Abstract

1. Needs grammar checks

2. Methods section requires review

3. Abstract is too long

4. Some statements in the results section do not belong to the results section

Introduction

1. Burden of disease broadly and well presented

2. However, the literature should be discussed to expose more knowledge mobilization backed by the burden of disease. There is need for spicing the introduction with more literature on knowledge mobilization backed by the burden of disease

3. Review and cite, where there is no citation, references are reasonably recent

Methods

1. A review of the methods in the abstract is required to align to the article methods

2. The methods section need strengthening and reorganization, including elimination of sub titles/themes

3. Inclusion and exclusion criteria require review and revision

4. Uniformity of search strategy is needed in consistency in the methods section and in the search strategy

5. The use of EndNote as a tool to clean duplicates is unclear , given that it is a citation tool

6. The data types are not clarified to justify the software used and data management as a whole

7. Introduction of quality assessment separately is unclear, since this ought to have been inbuilt in the design

Results

1. It is not clear, why a method used should arise in the results section

2. The mention of thematic analysis is an association with qualitative data to be found in the methods section as opposed to being discussed in the results section. Analysis in the results section poses a challenge to the use of excel sheets in the methods, implying quantitative data management. Review and clarify the issue clearly

3. The results are largely descriptive, though analytical level would have helped readers to clearly visualize and appreciate conclusively the value of knowledge mobilization.

4. Though results show thematic presentation, this was not clearly brought out in the methods section.

5. The thematic presented of results section, partially overlooked the quantitative element mentioned therein

6. The authors have concentrated on describing details in the selected articles instead of focusing more on the conclusions arising from knowledge mobilization of each paper - this would have supported the mention of theory of change

7. The articles do not quantify the numbers of articles, preferring to generalize the findings - for instance, numbers by articles, countries, regions, themes etc

Discussion

1. There is need to compare the findings and discuss the comparisons

2. Though findings are thematically discussed, the discussion is not aligned to the results

Recommendations

1. Too many, need collapsing and alignment to the findings and discussion

Conclusion

1. It is strangely placed after recommendations

2. It will require a revision after the corrections

General

1. Needs grammar checks

2. The article is too long

3. There is need to review and restructure the layout and structure of the article

4. Registration with Prospero is well done

5. The article requires economy of words, collapsing of themes and concepts, review and revision

6. I received and reviewed an already reviewed copy of the article, having raw track changes from another reviewer, instead of reviewing the authors’ original works. Was this an editorial error?

---

## [Editor Report]

Thank you for your thorough revisions to the manuscript entitled “Optimising knowledge mobilisation for mental health research in low-and-middle income countries: A systematic review of the state of knowledge and directions for future research” as previously requested by the reviewers.

The reviewers have noted that the paper has been substantially improved and is now well-refined. I agree with their assessment and am pleased to recommend your manuscript for publication.